# Sustainable Coffee Leaf Diagnosis: A Deep Knowledgeable Meta-Learning Approach

**Abdullah Ali Salamai** [1,*] and **Waleed Tawfiq Al-Nami** [2]

1   Department of Management, Applied College, Jazan University, Jazan 82822, Saudi Arabia
2   Department of Computer, Applied College, Jazan University, Jazan 82822, Saudi Arabia;
    walnami@jazanu.edu.sa
*   Correspondence: absalamai@jazanu.edu.sa

**Abstract:** Multi-task visual recognition plays a pivotal role in addressing the composite challenges encountered during the monitoring of crop health, pest infestations, and disease outbreaks in precision agriculture. Machine learning approaches have been revolutionizing the diagnosis of plant disease in recent years; however, they require a large amount of training data and suffer from limited generalizability for unseen data. This work introduces a novel knowledgeable meta-learning framework for the few-shot multi-task diagnosis of biotic stress in coffee leaves. A mixed vision transformer (MVT) learner is presented to generate mixed contextual attention maps from discriminatory latent representations between support and query images to give more emphasis to the biotic stress lesions in coffee leaves. Then, a knowledge distillation strategy is introduced to avoid disastrous forgetting phenomena during inner-loop training. An adaptive meta-training rule is designed to automatically update the parameters of the meta-learner according to the current task. The competitive results from exhaustive experimentations on public datasets demonstrate the superior performance of our approach over the traditional methods. This is not only restricted to enhancing the accuracy and efficiency of coffee leaf disease diagnosis but also contributes to reducing the environmental footprint through optimizing resource utilization and minimizing the need for chemical treatments, hence aligning with broader sustainability goals in agriculture.

**Keywords:** sustainable; coffee leaf diseases; leaf diagnosis; disease management; artificial intelligence; meta-learning

## 1. Introduction

Multi-task visual recognition is a revolutionary computer vision paradigm that allows training single artificial intelligence algorithms to perform many visual recognition tasks instantaneously. Different from single-task learning, in which independent models are trained for each separate task, resulting in redundant computation and enlarged complexity of the recognition algorithm, the multi-task paradigm makes use of shared representations across tasks, enabling the model to learn common designs and representations that are valuable for multiple tasks [1]. This method encourages the model to learn from its experiences with different tasks, thus enhancing its performance across the board. The model's efficiency and efficacy are both improved by the shared learning across tasks, and more comprehensive comprehension of the visual input is provided, allowing for the modeling of complex interdependencies and linkages. Better generalization is another benefit of multi-task visual recognition since the model is trained to extract high-level features that are robust and meaningful across different recognition tasks [2].

Deep learning (DL) has been revolutionizing the visual recognition tasks (i.e., classification, detection, segmentation, colorization, etc.) in many application domains owing to the powerful learning capabilities that enable the DL models to automatically process and extract patterns of disease from plant images without the need for any hand-crafted features or engineered features [3,4]. Among the DL methods, convolutional neural

networks (CNNs) are the dominating family of deep networks that have achieved many breakthroughs in different visual recognition tasks due to their capabilities for feature extraction stacked convolutional kernels. Recently, the vision transformers (ViTs) have evolved as state-of-the-art visual learning models that can model long-range dependencies, which enables outperforming the performance of CNNs on different vision tasks. However, their potential has still not been investigated for visual recognition tasks [5].

Knowledge distillation is a powerful technique in deep learning that enables model compression and facilitates the transfer of knowledge from a larger, complex teacher network to a smaller, more compact student network. The process involves training the student network to mimic the soft probabilities (logits) produced by the teacher network rather than learning from ground-truth hard labels [6]. By doing so, the student network learns not only from the teacher's final predictions but also from the rich intermediate information that the teacher encodes during training [7]. This distilled knowledge acts as a form of regularization, guiding the student network to focus on important patterns and generalize better to unseen data. Knowledge distillation has shown remarkable success in model compression, allowing the creation of more lightweight networks that retain the performance of their larger counterparts. In the context of multi-task visual recognition in agriculture, knowledge distillation can be particularly advantageous, as it allows the creation of efficient and accurate models capable of handling diverse tasks with reduced computational overhead [8].

Meta-learning, popularly recognized as "learning to learn," is a front-line learning method in machines that empowers the models to be able to adapt rapidly and successfully to new tasks with minimum training data [6]. Different from conventional learning methods, meta-learning algorithms can obtain more generalizable knowledge that could be straightforwardly applied to new and unseen tasks as they learn from a distribution of tasks rather than just one. This capacity is particularly useful in fields such as agriculture, where fluctuating weather patterns and new pests and diseases always present fresh recognition obstacles [7,8]. The integration of knowledge learned from several activities can empower meta-learners to learn a representation that is autonomous of the specific tasks being acted on. It can quickly adapt and learn from just a few samples or shots from the new activity since this meta-learner efficiently encodes knowledge about how to approach the assignment. Since the model becomes skilled at performing a variety of jobs and effectively learning from little data, meta-learning can contribute greatly to greater efficiency, speedier deployment, and better agility in the setting of multi-task visual recognition in agriculture [9,10].

The complexity of coffee farming and the breadth of tasks involved in leaf analysis make this an intriguing and challenging case study for multi-task visual recognition. The widespread crop loss caused by coffee leaf diseases is a major threat to agricultural production on multiple fronts, including yield and quality [11]. Pathogens, like fungi, bacteria, and viruses, cause these agricultural diseases by invading and damaging coffee plant tissues and organs. Hence, correct recognition of the types of these diseases, their causes, and their severity is all-important to implement effective management strategies to control the distribution of these diseases [12]. The four most important diseases that can impact coffee leaves are Cercospora leaf spot, brown leaf spot, rust, and leaf miner. The use of computer vision in precision agriculture has enabled the development of data-driven approaches with a continued emphasis on the early identification of crop losses from leaf diseases. Multi-task recognition of coffee leaf diseases in visual images is one of the many promising directions to improve disease diagnosis on smart farms [13]. Combining intelligent strategies for detecting coffee leaf disease can guarantee to safeguard long-term viability losses due to production declines resulting from coffee health reduction, thus ensuring continued superior quality satisfaction found in robust global demand for coffee products [6–12]. Moreover, it is vital to decide the severity of the stress impacting the coffee leaves, which is estimated by the proportion of the leaf surface that is wounded [8]. However, this multi-tasking activity is challenging, even for professionals, for many reasons,

such as the absence of exact delineation of the lesion; the variability of attributes between nodules of the same category (including color, outline, and magnitude); the occurrence of numerous lesions within the same leaf; the intra- and inter-observer variability, and the reality that various stresses could have similar effects [14]. Therefore, meta-learning and knowledge distillation are particularly applicable in this setting due to the complexity of coffee leaf analysis and the accompanying necessity for effective and adaptive learning strategies. The availability or practicality of the requisite labeled data and computer resources for training in traditional deep-learning approaches may be limited in agricultural contexts. Knowledge distillation allows us to efficiently transfer know-how, reducing the volume of data needed while simultaneously improving the model's replicability. In addition, meta-learning allows the model to learn from a variety of tasks and to adjust to novel tasks in the domain of coffee leaf cultivation [10–15].

### 1.1. Research Gaps

Despite the prevailing use of DL in a broad range of precision agriculture, however, these solutions usually require big datasets to be trained in an efficient manner, which is difficult to collect and annotate in real-world settings [11,12]. This, in turn, highlights the need for powerful hardware like graphical processing units (GPUs), which implies extra costs and expenses. This convergence of the above factors signifies the misalignment between the current DL solutions and the sustainability principles intrinsic to precision agriculture. Thus, it is clear that these solutions may not completely fulfill the sustainability requirements of precision agriculture, which is a great motive to further explore more resource-efficient and reasonably feasible alternatives [13]. Humans, conversely, can pick up new information and relocate fast based on a small number of examples, leading us to question whether intelligence gained from processing large datasets is the goal. Because of the low cost of a few samples of data, few-shot learning (FSL) has evolved as a paradigm to enable DL to learn from a small amount of data, and hence it is an interesting and potentially fruitful field of study for the detection of biotic stresses in coffee leaves [2]. Presently, transfer learning (TL), data augmentation, and meta-learning are the main three approaches to FSL. The purpose of TL is to acquire and apply information from one domain to another. Assuming there is adequate data in the source domain for learning, the learning algorithm would be fine-tuned by a few examples in the target domain to keep a satisfactory performance [3]. While TL solves the difficulties of model fragility and slow convergence by fine-tuning parameters, it still has some basic issues to deal with, such as the need for comparatively large datasets. For instance, the labeling and training of the classification model must be reperformed if multiple additional classification tasks are introduced. Intuitively, we can generate additional new instances or features with the help of data augmentation by rotating and scaling images, mixing them, oversampling, and other similar techniques. However, the performance improvement gained by data augmentation is still limited [4].

### 1.2. Novelty and Contribution

In response to the above-mentioned challenges, this study presents a new meta-learning framework for the sustainable diagnosis of coffee leaves through a few-shot estimation of biotic stresses and severity, with a powerful ability to generalize on unseen coffee data. The main contributions of this framework can be pointed out as follows:

- Motivated by the ability of ViTs to learn long-term dependencies, a mixed vision transformer (MVT) is introduced to simultaneously extract rich contextual representations from both support and query images rather than extracting visual leaf features independently from the query and support inputs. In the MVT layer, we introduced contextual attention to capturing the significant information regarding the biotic stress and its severity degree in a way that enables perfect alignment of visual representations from both the query and support set.

- A new meta-training strategy is presented to empower the meta-learner to adapt its learnable according to the underlying task in a way that improves the generalizability of unseen data.
- A novel knowledge distillation is introduced in our framework to keep a balance between learning new knowledge and preservation of previously acquired knowledge in a way that removes the appalling forgetting problem.
- Exhaustive proof-of-concept experiments are conducted to evaluate and compare the performance of the proposed framework against the state-of-the-art methods on a case study of coffee leaf datasets. The findings demonstrate the proposed is qualified to be adopted as an efficient tool for delivering insightful diagnoses for coffee diseases in smart farming environments.

### 1.3. Outline

The overall structure of this work is composed of six primary sections. Section 2 reviews the related literature. Section 3 debates the methodology of the proposed framework. The experimental settings of this work are given in Section 4. Section 5 presents the experimental results and the corresponding discussion. Finally, the concluding remarks of this work are provided in Section 6.

## 2. Literature Review

This section provides an in-depth analysis of the related literature with a primary emphasis on visual recognition studies for the detection of leaf diseases in coffee plants. Two branches of studies are discovered in this spectrum, as described in the following subsections.

### 2.1. DL-Based Visual Recognition

Research on multi-task visual recognition has been gaining increasing interest in recent years. However, it is still in its infancy stage. For example, Esgario et al. [5] explored the potential of CNNs for multi-task diagnosis of biotic stress from images of coffee leaves by classifying biotic stress and estimating their severity. Similarly, Putra et al. [6] applied a CNN and compared its performance against common models, such as LeNet, AlexNet, ResNet-50, and GoogleNet, for classifying between Arabica and Robusta coffee plants. In [7], a segmentation of the symptomatology was carried out using a threshold-based method to create a system that can classify the individual symptoms of coffee leaves. Manual features were used to categorize the symptoms into leaf miner and rust groups. The severity of the symptoms could be measured, and specific symptoms could be identified, thanks to the lesion segmentation findings. Despite the impressive outcomes, this method of segmentation is highly susceptible to environmental factors, like lighting changes and specular reflection, because it ignores the positional relation between the pixels. Novtahaning et al. [8] introduced an ensemble DL system that exploits the notion of TL to integrate multiple pre-trained CNNs to extract the disease patterns from the in-field images of coffee leaves. Then, the most powerful feature extractor is nominated to dominate the ensemble architecture, and its features are passed to the final classification head. Additionally, Esgario et al. [9] introduced a multi-step approach that includes a semantic segmentation step followed by a severity estimating step for detecting diseases and pests of coffee leaves. U-net architecture is used in the earlier step, while a CNN was used to implement the later step. These two steps were evaluated disjointedly to shed light on the optimistic and undesirable arguments of each one. To aid professionals and farmers in identifying and quantifying biotic stresses utilizing pictures of coffee leaves captured by mobile phones, a custom app was developed and deployed on the Android platform, including segmentation, severity computation, and symptom categorization. Similarly, Tassis et al. [10] introduced a three-stage DL system that combined different CNNs to systematize the detection of lesions presented in mobile-captured in-field imageries encompassing fragments of the coffee tree. The first stage applied Mask R-CNN to perform instance segmentation for coffee leaves. Then, the UNet and PSPNet models were applied

to perform semantic segmentation at the second stage, and lastly, residual convolution networks were adopted to perform classification. In addition, Yamashita and Leite [11] emphasized developing a lightweight CNN, based on the MobileNet model, to detect coffee leaf disease on edge computing devices namely low-cost microcontroller board. It demonstrated the practicability of using CNNs for detection of diseases in coffee leaf images in resource-constrained devices with no internet connection.

A prominent and common downside for the above methods is the data-hungry nature that makes them unable to learn well from small-size data, which is a common case for coffee biotic stress. This, in turn, implies the weak generalizability of these methods for an unseen dataset.

### 2.2. Meta-Learning in Visual Recognition

In response to the challenges of learning from small-size data, the research community moved toward meta-learning approaches. For instance, Li and Yang [12] introduced a few-shot learning framework to identify cotton pests using only a few images, in which two datasets were used to validate the efficiency and viability of the model. This framework trained the parameters of CNN as a feature extractor according to a triplet loss function to help differentiate various pest species to guarantee the system toughness. Not only that, but the compiled CNN circuit in the FPGA and the control program in the ARM were both successfully used to obtain the model functioning in an embedded gate. In addition, Hu et al. [13], introduced a low-shot learning approach to disorder recognition in tea leaves to facilitate timely prevention and control measures. Disease regions on tea leaf images were segmented utilizing the support vector machine (SVM) technique, with color and texture features extracted. By feeding segmented disease spot images into an enhanced conditional deep convolutional generative adversarial network (C-DCGAN) for data augmentation, a VGG16 model is taught to recognize diseases in tea leaves. In [15], Li and Chao proposed a semi-supervised few-shot learning strategy for identifying plant leaf diseases, in which a subset of the publicly available PlantVillage dataset is used, and it is further subdivided into a source domain and an end-user domain. The experiments were conducted taking into account the domain separation and few-shot parameters (*N*-way, *k*-shot) to verify the rightness and generalization of developed semi-supervised few-shot methodologies. To flexibly choose the amount of unlabeled data for pseudo-labeling in the semi-supervised process, a confidence interval was used in this work. Moreover, Tseng et al. [14] looked into how model-agnostic meta-learning (MAML) can be used to gain insight from a wide variety of global datasets and boost results in areas with limited available information. The findings of this work stated that MAML outperforms pre-trained and random initial weights in a wide range of tasks and countries (including Togo, Kenya, and Brazil). It also investigated MAML's potential benefits across a variety of target data size commands. Across a broad range of training set sizes and positive-to-negative label ratios, the findings supported that MAML performs better than alternative meta-learners, suggesting its general applicability for territory usage multiple crop mapping.

Despite the research efforts devoted to the few-shot computer vision applications, this family of solutions is still in its infancy and has not been deeply investigated in visual recognition tasks. This implies the promise of this research direction for the detection of biotic stresses in coffee leaves. Thus, in this work, we seek to fill this gap by studying meta-learning for improving the multi-task diagnosis of coffee leaf disease under a data scarcity scenario. To this end, this work presents a multi-task meta-learning framework as an important step towards flexible and efficient solutions to problems in precision agriculture that require a wide range of resources. The design principles of the proposed model validate remarkable flexibility in handling emerging tasks and obstacles, radically cutting down on the time and effort spent retraining and freeing up precious computational resources in the process. With the adaptability and multitasking abilities of our model, the process of coffee disease management may be directed with more efficiency, which makes

it an asset in the context of changing agricultural dynamics, where a quick response to new issues is of utmost importance.

## 3. Methodological Framework

### 3.1. Problem Formulation

The multi-task classification of coffee leaves is designated as a few-shot learning dilemma, whereby a set of labeled coffee leaf images is given as a training set, and a set of labeled coffee leaf images is given as a testing set. Both sets share non-overlapping label space. Therefore, given a few samples from the labeled set; our objective is to learn a model that could generalize unseen coffee leaf images from the unlabeled set. To achieve this objective, meta-learning with episodic training could be applied to fine-tune the generalizability by imitating the limited data scenario met during the inference through generating stable episodes. An episode is shaped with two subsets: one contains a few labeled coffee leaf images for learning, and the other one contains labeled images, in which the labels are utilized to compute the error rate of the model's prediction in each episode. In addition, the "*N*-way, *k*-shot" task is defined by every training episode, with *N* pointing to the number of biotic/severity classes in each episode, while *k* denotes the number of images in each class. The problem of multi-task diagnosis of coffee leaves is formulated as a generic meta-setup known as *N*-way, *k*-shot classification.

This formulation is built up with three phases described below:

1.  *Meta-training* phase: the multi-task learner is initiated to learn from a set of meta-training data.
2.  *Meta-validation* phase: the meta-learner leverages meta-validation data to assess its classification performance on unobserved tasks that are not encountered during the meta-training phase. This assessment estimates the meta-generalizability of the trained learner and provides feedback to update and fine-tune the parameters of the meta-learner.
3.  *Meta-testing phase:* different from the previous phase, the meta-testing data are exploited here to evaluate the final performance of the meta-learner for both biotic stress classification and severity estimation.

In the first phase, a big enough coffee dataset is used to *train* the multi-task learner, $\mathcal{A}$, over a finite number of episodes. For each episode, the data $\mathcal{T}_j = \left( \mathcal{D}_{\mathcal{T}_j}^{tr}, \mathcal{D}_{\mathcal{T}_j}^{val} \right)$ are attained through sampling coffee leaf images $(xx_i, y_i)$ from the complete dataset $\mathcal{D}$. The tuple $\left( \mathcal{D}_{\mathcal{T}_j}^{tr}, \mathcal{D}_{\mathcal{T}_j}^{val} \right)$ describes a training and a validation set of an episode, in which each set comprises a small number of coffee leaf images. According to the *N*-way, *k*-shot definition, the process of the data sampling process poses that the training set $\mathcal{D}_{\mathcal{T}_j}^{tr}$ comprise precisely *N* classes with *k* images belonging to each class, suggesting that $\left| D_{\mathcal{T}_j}^{tr} \right| = N \times k$. Additionally, the factual labels of images in the validation set $\mathcal{D}_{\mathcal{T}_j}^{val}$ need to exist in the train set $\mathcal{D}_{\mathcal{T}_j}^{tr}$ of a certain episode $\mathcal{T}_j$. The training set $\mathcal{D}_{\mathcal{T}_j}^{tr} = \left\{ (x_s, y_s) \right\}_{s=1}^{S}$ is also a support set, because it figuratively supports the learner's decisions on the validation set $\mathcal{D}_{\mathcal{T}_j}^{val} = \left\{ (x_q, y_q) \right\}_{q=1}^{Q}$ known as query set. The multi-task learner, $\mathcal{A}$, can be defined as $y_* = f_\theta(x_*)$, with $*$ signifying s or q, and is prone to high variance because of the large dimensions of $x_*$. Thus, coffee leaf images in both support and query sets are encoded into latent space via an embedding function $\Phi_* = f\phi(x_*)$. Supposing that the embedding function is static throughout the training of $\mathcal{A}$ on each episode, then the multi-task learner seeks to optimize the following objective:

$$\theta = \mathcal{A}\left( \mathcal{D}_{\mathcal{T}_j}^{tr}; \phi \right) = \arg\min_{\theta} \mathcal{L}^{\text{base}}\left( \mathcal{D}_{\mathcal{T}_j}^{tr}; \theta, \phi \right) + \mathcal{R}(\theta), \tag{1}$$

The terms $\mathcal{L}^{\text{base}}$ and $\mathcal{R}$ denote cost function and regularization factor, respectively. Here, $\phi$ stipulates expectations regarding 'how to learn', e.g., the optimizer. The goal of our meta-learning coffee leaf classifier is to train a decent embedding function, with the aim of reducing the average loss of the multi-task learner on the query set. Officially,

$$\phi = \underset{\phi}{\arg min}\, \mathbb{E}_{\mathcal{T}}\left[\mathcal{L}^{\text{meta}}\left(\mathcal{D}^{\text{val}};\theta,\phi\right)\right] \tag{2}$$

By the completion of meta-training, the learner's performance is evaluated on a meta-testing set $\mathcal{S} = \left\{\left(\mathcal{D}_j^{tr},\mathcal{D}_j^{ts}\right)\right\}_{j=1}^{J}$. The assessment is performed using the distribution of the test coffee leaf images, as follows:

$$\mathbb{E}_{\mathcal{S}}\left[\mathcal{L}^{meta}\left(\mathcal{D}^{ts};\theta,\phi\right), \text{with}\theta = A\left(\mathcal{D}^{tr};\phi\right)\right]. \tag{3}$$

### 3.2. Mixed Vision Transformer (MVT)

This section argues for the design of our multi-task learner backbone that seeks to handle the challenge of generalizability on unseen types of biotic stresses from the interactive standpoint. Figure 1 displays the construction of our multi-task learner, comprising three chief learnable modules and dual classification heads. The design of our MVT extractor consists of a stack of three MVT blocks for multiparty contextual representation learning of the support and query images. At the early block of the backbone, there is a batch of support images, $X_s \in \mathbb{R}^{B_s \times H_{x_s} \times w_{x_s} \times 3}$, and one query image, $X_q \in \mathbb{R}^{B_q \times H_{x_q} \times w_{x_q} \times 3}$. Inspired by ViT [16], the proposed MVT block is composed of the contextual multi-head contextual attention, two fully connected layers (FCLs) with residual connectivity, LayerNorm (LN), as well as Gaussian error linear unit (GELU) activation. Also, the design of the backbone feature extractor follows the non-hierarchical design, in which the MVT block shares the same learnable parameters, embedding and manipulating feature maps with shared dimensions. In this case, our model handles the adjustable extent of the token sequence.

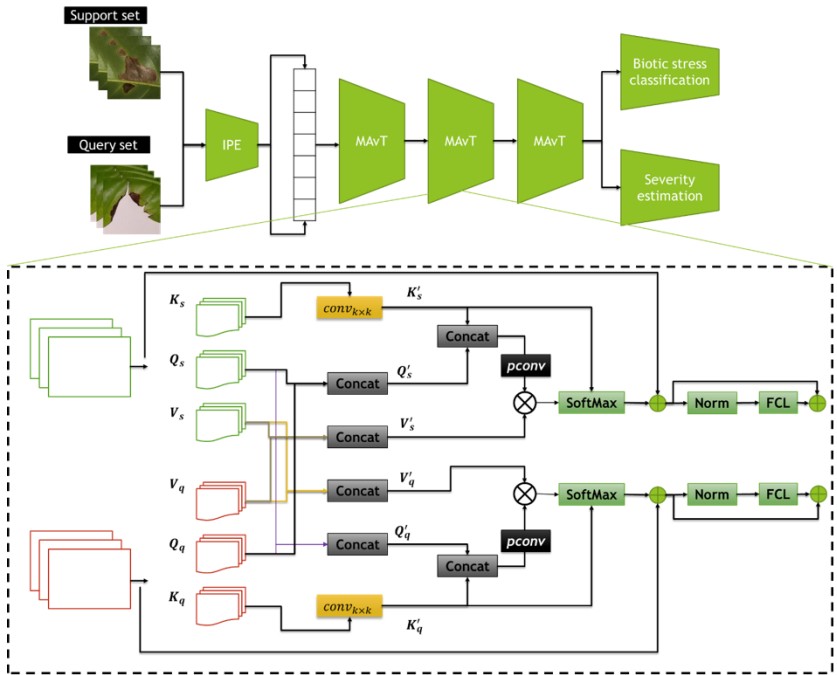

**Figure 1.** Systematic diagram of the architecture of the proposed meta-learning approach for multi-task few-shot classification of coffee leaves.

The next MVT blocks share a comparable structure and calculate representational maps through the gradual reduction in sequence lengths and expansion of the number of channels. In standard ViT, a patching operation was applied to split the raw images into a group of

non-overlapping patches. Next, a linear embedding layer is applied to flatten the generated patches into $C_1$ dimensions. Then, positional embedding is generated for those patches before being passed into the ViT layer. However, traditional patch embedding suffers from discontinuity that limits the ability to model local features. To overcome this limitation, an interactive patch embedding (IPE) block is built up by stacking multiple strided convolution layers of $k = 3 \times 3$ convolution, switch normalization (SN) [17], and GELU. The patch size is computed as $P = 2k$. This design enables conserving significant low-level features and local spatial information in coffee leaf images because of the injection of inductive biases by convolutions. Moreover, the IPE block empowers the MVT to have more elasticity over original ViTs, by alleviating the constraints such that the input dimensions (whether in query or support set) are firmly dividable by the predefined patch size. The IPE uses two affine transformations, $\aleph$, prior to and following the stacked convolutions for rescaling and shifting the image feature, hence making the learning performance steadier on small datasets, which is common for the coffee leaf dataset. The pipeline of IPE can be expressed as follows (Algorithm 1):

---

**Algorithm 1:** Interactive Patch Embedding (IPE)

---

1    **IPE(x):**
2    $\aleph(x) = \text{Diag}(\alpha)x + \beta$
3    For $i = 1, \dots, k$ do:
4        $F_i(x) = \text{GELU}(\text{SN}(\text{Conv}_{3 \times 3}(x)))$
5    $x' = \aleph(F_k(\dots (F_2(F_1(\aleph(x))))))$
6    $x'' = \text{Reshape}(x')$
7    **Return** $x''$

---

In our MVT block, the IPE is applied to generate embedding for both support and query images as follows.

$$\begin{aligned} X'_s &= IPE(X_s) \\ X'_q &= IPE(X_q) \end{aligned} \tag{4}$$

For each attention head, $(h = 1, \cdots, H)$, the embedding of support $X'_s$ and $X'_q$ are mapped to $Q^i_s$, $K^i_s$, $V^i_s$ and $Q^i_q$, $K^i_q$, $V^i_q$, respectively. To maintain lightweight computation for our attention, a strided depth-wise separable convolution (dwconv) is introduced to promote the subsampling of visual representation maps for matrices K and V.

$$Q^i_q = X'_q W^i_Q, \& Q^i_s = X'_s W^i_Q \tag{5}$$

$$K^i_q = dwconv_{str=2}\left(X'_q\right) W^i_K, \& K^i_s = dwconv_{str=2}\left(X'_s\right) W^i_K \tag{6}$$

$$V^i_q = dwconv_{str=2}\left(X'_q\right) W^i_V, \& V^i_s = dwconv_{str=2}\left(X'_s\right) W^i_V \tag{7}$$

In the above formula, $W^i_Q \in \mathbb{R}^{C_1 \times d_h}$, $W^i_K \in \mathbb{R}^{C_1 \times d_h}$, and $W^i_V \in \mathbb{R}^{C_1 \times d_h}$ denote the trainable parameters shared for the projection layer for both input sets. $d_h = \frac{C_1}{H}$ denotes the dimension of the projected features.

In our case, the size of the support batch varies from the size of the query batch. Also, biotic detection or/and severity assessment is performed for each query instance disjointedly, since query imageries are unrelated, and thereby the recognition of each is independent of others. The simple way to address the variation in sizes is to direct a pair of images (one from query and one from support) at each time; however, replicating this procedure for each image in the support set leads to high time complexity. Thus, we propose enabling contextual attention to consider the attention between the query image and all images belonging to the same class in the support set. The contextual attention mechanism is introduced into the MVT layer to fuse the key-value information related to query and support input attention. To fuse the key-value information from the support-

related feature maps to the query-related feature maps and fuse the key-value information from the query-related feature maps to the support-related feature maps, a pointwise convolution is applied to update the channel dimensions of maps to be matched and then the key-value information of both inputs is concatenated.

$$\widetilde{Q}_q^i = \left\| K_q^i, \frac{1}{B_s} \sum_{B_s} Q_s^i, \ \widetilde{V}_q^i = \right\| V_q^i, \frac{1}{B_s} \sum_{B_s} V_s^i \tag{8}$$

$$\widetilde{Q}_s^i = \left\| K_s^i, \frac{1}{B_q} \sum_{B_q} Q_q^i, \ \widetilde{V}_s^i = \right\| V_s^i, \frac{1}{B_q} \sum_{B_q} V_q^i \tag{9}$$

The term $\|\cdot$ represents the concatenation layer. To sum up, the computation of multihead contextual attention can be summarized as follows:

$$X_q'' = \text{Concat}\left( h_q^1, \dots, h_q^H \right) W_O \tag{10}$$

$$h_q^i = \text{Attention}\left( \widetilde{Q}_q^i, K_q^i, \widetilde{V}_q^i \right) \tag{11}$$

$$X_s'' = \text{Concat}\left( h_s^1, \dots, h_s^h \right) W_O \tag{12}$$

$$h_s^i = \text{Attention}\left( \widetilde{Q}_q^i, K_q^i, \widetilde{V}_q^i \right) \tag{13}$$

Next, the linear layers are applied to each patch with robust feature representations, while the residual connection is used to improve the gradient flow.

$$\begin{aligned} X_q''' &= Linear\left( \text{LN}\left( X_q'' \right) + X_q'' \right) \\ X_s''' &= Linear\left( \text{LN}\left( X_s'' \right) + X_s'' \right) \end{aligned} \tag{14}$$

*3.3. Adaptive Meta-Training*

The primary objective of meta-training is to allow the multi-task learner to perfectly train an embedding function, $f_\phi$, to be able to generalize well on new, unseen coffee data. Instead of rebuilding and/or retraining new learners to learn the embedding function, our framework suggests that the pre-training of vision learners on single- or multi-task data is favorable for empowering the embeddings of the underlying multi-task learner. Thus, the meta-training data were combined from all episodes according to the following formula:

$$\mathcal{D}^{combine} = \left\{ (\mathbf{x}_i, y_i) \right\}_{k=1}^K = \cup \left\{ \mathcal{D}_1^{tr}, \dots, \mathcal{D}_i^{tr}, \dots, \mathcal{D}_I^{tr} \right\}, \tag{15}$$

Next, the embedding function can be defined as follows:

$$\phi = \underset{\phi}{\arg min} \mathbb{E}_{\mathcal{T}} \left[ \mathcal{L}^{\text{cf}} \left( \mathcal{D}^{\text{combine}}; \theta, \phi \right) \right] \tag{16}$$

where the $\mathcal{L}^{\text{cf}}$ represent the categorical focal loss [18] between the learner's predictions and coffee leaves labels.

Different from early meta-learning mechanisms, the proposed framework seeks to optimize an adaptive cost function to enforce the inner-loop optimization process to achieve high generalizability for new episodes during the training. To this end, an adaptive cost function $L\phi(\cdot)$ is presented and updated with a small learner with trainable parameters, $\phi$. This way, the inner-loop update can be formulated as follows:

$$\theta_{i,j+1} = \theta_{i,j} - \alpha \nabla_{\theta_{i,j}} \mathcal{L}_\phi \left( \tau_{i,j} \right) \tag{17}$$

The term $\tau_{i,j}$ represents the state of the episode $\mathcal{T}_i$ at time-step $j$, which is always defined by the training set in the conventional meta-learning paradigm. Various episodes might require including some regularization factors or secondary cost functions throughout training to generalize well. Unlike gradient descent adaptation, which exhibits high computing complexity, we proposed to take advantage of deformable transformation for the creation of adaptive feature interactions, thereby making the updates of the inner loop more adaptive. This can be expressed as follows:

$$\phi' = \gamma\phi + \pi \tag{18}$$

where $\phi$ symbolizes the parameters of the multi-task learner and $\gamma$ denotes $\pi$ the deformation parameters made by the learner $g(\tau j ; \psi)$ with parameter $\psi$. The meta-training process seeks to enable the multi-task learner to generalize across diverse episodes through optimizing the parameters $\theta$, $\phi$, *and* $\psi$, and then we perform outer-loop optimization for episode $\mathcal{T}_i$ using their own learner and query samples as formulated below:

$$(\theta, \phi, \psi) \leftarrow (\theta, \phi, \psi) - \eta \nabla_{(\theta, \phi, \psi)} \sum_{\tau} \mathcal{L}\left(\mathcal{D}_i^{ts}; \theta_i\right) \tag{19}$$

Knowledge distillation [19] is a popular technique for knowledge transmission from a robust teacher–learner to a minor student learner. At each step of the model's learning process, a distillation loss function is required to make sure that knowledge retaining and acquisition are balanced in the best way possible. Rather than directly applying the embedding function on the meta-testing set, the learned knowledge in the embedding function is distilled into a new embedding function sharing the same building structure. Then, we train the new model with a set of parameters $\phi'$ to optimize dual-task focal loss and the Jensen–Shannon (JS) divergence between outputs and soft targets:

$$\phi' = \underset{\phi'}{\arg min}\left(\alpha \mathcal{L}^{cf}\left(\mathcal{D}^{combine}; \phi'\right) + \beta\, JS\left(f\left(\mathcal{D}^{combine}; \phi'\right), f\left(\mathcal{D}^{combine}; \phi\right)\right)\right) \tag{20}$$

where usually $\beta = 1 - \alpha$.

To evade the problem of disastrous forgetting in our framework, the above distillation loss is integrated as part of our loss function to empower the learner of the current episode to keep achieving good performance similar to the learners' performance in the previous episodes. The JS distillation is attributed to its ability to deal with probability distributions containing extreme values which is often the case in multi-task learning scenarios, in which there is a high need to balance the contributions of different tasks, mitigating the impact of outliers or noisy data. In addition, JS divergence possesses the desirable properties of symmetry and smoothness, which guarantees that the order of the input distributions does not affect the result. JS divergence can be interpreted as a softened form of discrimination between probability distributions. In our design, we follow the approach from Born again [20] to put on JS successively to engender manifold generations. For each episode, the knowledge of the embedding function at the $k - th$ generation is transmitted to the embedding function of the next generation:

$$\phi_k = \underset{\phi}{\arg min}\left(\alpha \mathcal{L}^{cf}\left(\mathcal{D}^{combine}; \phi\right) + \beta JS\left(f\left(\mathcal{D}^{combine}; \phi\right), f\left(\mathcal{D}^{combine}; \phi_k\right)\right)\right) \tag{21}$$

This operation is repeated for $K$ times, and the $\phi_K$ is designated as an embedding function to learn discriminative features.

For each meta-testing tuple $\left(\mathcal{D}_j^{tr}, \mathcal{D}_j^{ts}\right)$, the multi-task learner is instantiated as Bayesian multivariate logistic regression with weight $W$ and bias parameters, $b$, as defined below:

$$\theta = \underset{\{W,b\}}{\arg min} \sum_{t=1}^{T} \mathcal{L}_t^{cf}\left(W f_{\phi}(x_t) + b, y_t\right) + \mathcal{R}(W, b) \tag{22}$$

## 4. Experimental Materials and Setups

This section provides an in-depth discussion of the experimentation of this work in terms of the coffee data adopted to evaluate the proposed model, the competing baselines with which our model is compared, the evaluation indicators used in experiments, and the implementation setups.

### 4.1. Materials

In this work, we train and evaluate the proposed model on the multi-task biotic stresses dataset from [5]. It contains images of Arabica coffee leaves exaggerated by the core biotic stresses that influence the coffee tree, which was captured using different smartphones at diverse times of the year in the state of Espírito Santo, Brazil. The photos were taken from the abaxial (lower) side of the leaves under partially controlled conditions and placed on a white background. In addition to that, in-field images were not used due to the occlusion of leaf parts and their inclination concerning the camera, making the disease severity estimation infeasible in several cases. The images were captured from the underside, or abaxial side, of the leaves and then superimposed onto a white background for viewing. In addition, in-field imagery was not leveraged because of the obstruction of leaf slices and their predisposition relative to the camera, which makes the estimate of disease severity impracticable in some cases. There were no criteria used during image collection, which resulted in a more diverse dataset. Leaf miner, brown leaf spot, rust, and Cercospora leaf spot were just some of the biotic challenges experienced by the Arabica coffee plants that led to the collection of 1747 photos of healthy and diseased leaves. An expert used the acquired photos to perform the procedure of biotic stress identification for dataset labeling. Two datasets were created from the collected pictures: symptom-only pictures and the original, full-leaf images from which they were extracted. The following paragraphs provide descriptions of each dataset.

- **Leaf dataset**: There are complete-leaf photos from the original source, and they have been annotated with information about the most common biotic stress and its intensity. There were 372 photos showing leaves experiencing multiple stressors, of which 62 indicated stresses of similar severity. Given that this study can only identify a single stress per leaf, multi-stress categorization is outside the purview of this work. Since it is challenging to visually determine which stress is predominant, these 62 photos with equal severity were not used on this dataset. The severity of stress was determined by applying automatic image processing techniques, such as a mask, to isolate symptoms and identify which leaves are stressed. All picture segmentation findings were manually checked for accuracy. Images with inadequate segmentation were removed from the severity calculation and were examined manually by a trained professional using visual estimate techniques. This led to five degrees of severity, namely healthy, very low, low, high, and very high.
- **Symptom dataset**: This dataset was produced by selectively cropping the original photos so that just a single stress was visible in each. The cropping process involved 2147 photos of symptoms.

To make things simpler, for both tasks, we provide a detailed description of the class distribution of the data, presented in Table 1. Moreover, samples for different types of coffee leaves from the above datasets are provided in Figure 2.

**Table 1.** Summary of class distribution of the dataset.

| Biotic Stress | Healthy | Leaf Miner | Rust | Brown Leaf Spot | Cercospora Leaf Spot | Total |
|---|---|---|---|---|---|---|
| Leaf dataset | 272 | 387 | 531 | 348 | 147 | 1685 |
| Symptom dataset | 256 | 593 | 991 | 504 | 378 | 2722 |
| Severity | Healthy | Very low | Low | High | Very high | / |
| Leaf dataset | 272 | 924 | 332 | 101 | 56 | 1685 |

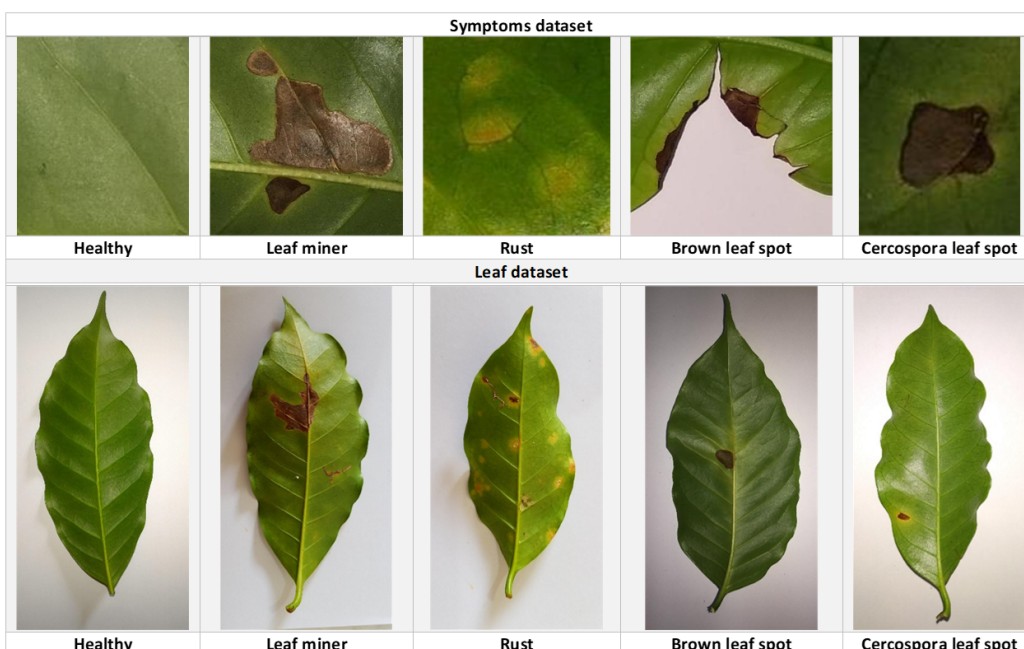

**Figure 2.** Visualization of samples from different classes in coffee leaf dataset and coffee symptoms dataset.

### 4.2. Baselines

The evaluation of the proposed method should involve a comparison with the state-of-the-art meta-learning algorithms. To this end, this experimental part of this work used six meta-learning methods as our baselines, namely SiameseNet [21], TripletNet [22], ProtoNet [23], MatchingNet [24], MAML [25], and RelationNet [26]. For each baseline, three vision transformers are applied as a backbone, namely MobileViTv2 [27], LeViT [28], and Mobile-Former [29].

### 4.3. Evaluation Metrics

The evaluation of the performance of deep learning models for both classification and severity estimations is designated using a different set of metrics. Four of them (i.e., *accuracy*, *precision*, *recall*, *F1-measure*) are calculated according to the confusion matrix comprising true positive (*TP*), true negative (*TN*), false positive (*FP*), and false negatives (*FN*). These metrics are computed as follows:

$$Accuracy = \frac{TP + TN}{TP + TN + FP + FN} \tag{23}$$

$$Precision = \frac{TP}{TP + FP} \tag{24}$$

$$Recall = \frac{TP}{TP + FN} \tag{25}$$

$$F1 - measure = 2 * \frac{Recall \times Precision}{Recall + Precision} \tag{26}$$

The area under the receiver operating characteristic (AUROC) curve [30] is also reported as a numerical metric for model evaluation in both tasks.

### 4.4. Environmental Setups

The implementation of deep learning models is coded with the Pytorch library running on Python 3.8.0 environment. All experimental tools are fixed on a Dell workstation equipped with 128 GB RAM, Intel® Xeon® Silver 4316 Processor (30 M Cache, 2.30 GHz),

and operated with Windows 10 64-bit OS. The learning of the models is accelerated with Nvidia GeForce RTX 3090. To guarantee consistent comparative results, experiments are run with standardized hyper-parameter settings. The specifics of model hyper-parameters are presented in Table 2.

**Table 2.** Summary of training hyper-parameters in our benchmark.

| Hyper-Parameters | Assigned Value |
| --- | --- |
| Batch Size | 32 |
| # Epochs | 80 |
| Input size | $244 \times 244 \times 3$ |
| Patch size | 16 |
| Optimizers | AdamW |
| Loss function | Focal loss |
| # Attention heads | 8 |
| Learning rate | 0.001 |
| Decay | 0.0003 |
| Kernel | $3 \times 3$ |
| Linear | 128 |
| Dropout rate | 0.3 |

## 5. Results and Discussion

To determine how well the proposed models fit the challenges of biotic stress identification and severity estimate, a different set of experiments was carried out in this section. The findings are reported in the following parts, for both datasets (coffee leaves and symptoms).

### 5.1. Comparative Analysis

In this section, we provide a detailed discussion of the numerical analysis of the experimental comparisons between the proposed method and competing baselines. For each dataset, we report the performance of the models under five-way one-shot and under five-way five-shot scenarios. The experimental conditions are kept constant for all experiments to maintain the fairness of the comparisons. The results of comparative experiments are reported in terms of the average and standard deviation of the results from training data folds. In Table 3, we present the quantitative results obtained from the different methods with five-way five-shot training on the leaf dataset. The tabulated results represent the performance of the models on coffee biotic stress classification as well as the severity estimation tasks. Notably, the SiameseNet achieves the lowest stress classification accuracy (ranging from 91.73% to 92.49%) for all backbones, and it also achieves the lowest severity estimation accuracy (ranging from 88.38% to 90.55%) for all backbones. Comparatively, the TripletNet can achieve improved stress classification accuracy but achieve similar severity estimation accuracy except for Mobile-Former (with 92.38% accuracy). This reflects that the appropriate selection of backbone network is important to severity estimation. In addition, ProtoNet can notably achieve higher results than TripletNet across all performance indicators for both tasks. Similar performance can be observed for MatchingNet with biotic stress classification accuracy in the range from/to and severity estimation accuracy in the range from/to. The performance of RelationNet is similar to TripletNet in the biotic stress classification task; however, this is not the case for the severity estimation task. Further, it is worth noting that the NAML achieved the highest performance (with accuracy between 96.15 and 96.69), overcoming the other baselines for both tasks. This reflects the advantage of combining optimization-based meta-learning into FSL for improving the generalizability of the underlying model. More importantly, the proposed model is achieving remarkable improvement over all baselines reflecting the ability of our model to learn the discriminatory features necessary for biotic stress classification in coffee leaves and simultaneously to learn the severity attributes from small data scenarios.

To further interpret and analyze the performance of the proposed model (five-way five-shot), Figure 3 provides a visualization of the class-level performance by plotting the confusion matrices for both classification and severity estimation tasks. For coffee biotic stress classification, it is worth noting that the class "rust" attains the lowest detection performance with 97.2% precision, while the classes "healthy and Brown leaf spot and Cercospora leaf spot" attain the highest detection performance with 100% precision. For severity estimation, it is worth noting that the class "very high" attains the lowest detection performance with 90% precision, while the class "very low" attains the highest detection performance with 98.2% precision. These findings coincide with the findings from previous studies [2,5], where the lowest performance was attained in this same class. This can be attributed to the high resemblance between lesions in these classes. However, different from the previous studies [2,5], the proposed model can recognize the different types of stresses with high confidence, which further demonstrates the discriminatory power of our model.

Using the leaf dataset for five-way one-shot training, the quantitative results of biotic stress categorization using various approaches are shown in Table 4. The results demonstrated the effectiveness of the models in classifying and estimating the severity of biotic stresses on coffee. SiameseNet uses Mobile-Former as its backbone to achieve the lowest stress classification accuracy (89.48%) and the lowest severity estimation accuracy (87.8%). While TripletNet outperforms the other backbones and SiameseNet by a small margin (93.5%) when trained using LeViT for stress classification, the other backbones perform about the same. Based on these findings, it appears that selecting the right backbone network is crucial for the multi-task diagnosis of coffee leaves. Furthermore, the ProtoNet performs better than the TripletNet on both tasks across the board. It is also noteworthy that the NAML, MatchingNet, and RelationNet all outperform the other baselines by around the same margins on both tasks. Our model's remarkable ability to learn multi-task features allows it to significantly beat all baselines. Although it was expected that model performance would suffer when going from five shots to one, the findings show that the suggested model still does well in a five-way one-shot scenario.

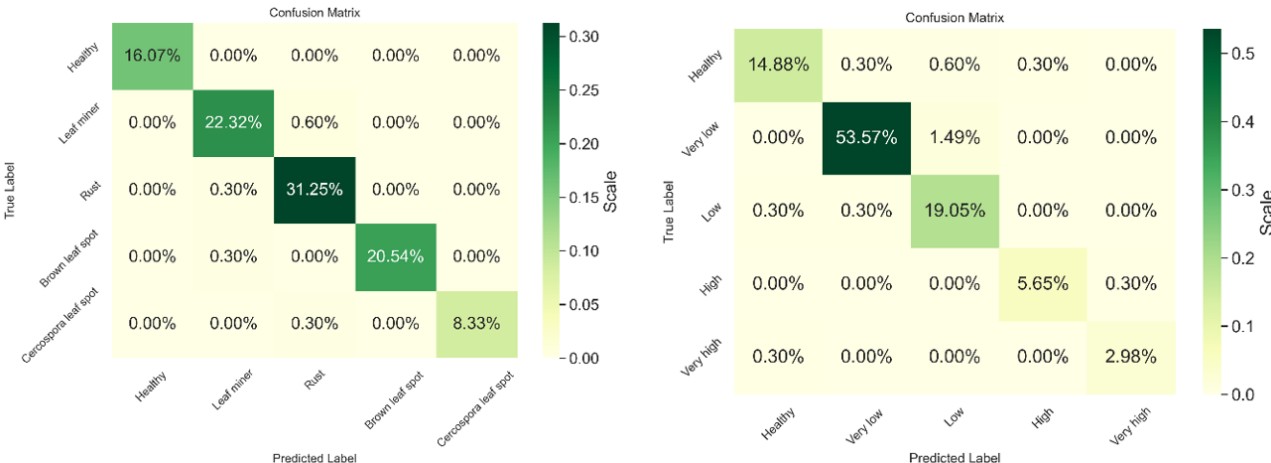

**Figure 3.** Confusion matrix of the proposed model on leaf dataset for coffee biotic stress classification (**Left**) and severity estimation (**Right**) in 5-way 1-shot setting.

**Table 3.** Comparison of numerical results of the proposed model against baselines on leaf dataset in 5-way 5-shot setting.

| FSL Method | Backbone | Biotic Stress Classification | | | | | Severity Estimation | | | | |
|---|---|---|---|---|---|---|---|---|---|---|---|
| | | Accuracy | Precision | Recall | F1-Score | AUC | Accuracy | Precision | Recall | F1-Score | AUC |
| SiameseNet | MobileViTv2 | 92.34 ± 5.22 | 92.88 ± 3.10 | 92.48 ± 3.77 | 92.68 ± 3.89 | 95.61 ± 2.79 | 88.38 ± 2.72 | 89.06 5.96 | 90.68 3.48 | 89.86 3.74 | 93.09 ± 6.00 |
| | LeViT | 92.49 ± 5.22 | 91.49 ± 2.59 | 92.3 ± 2.23 | 91.9 ± 3.46 | 96.55 ± 2.28 | 90.55 ± 2.27 | 87.79 ± 5.13 | 91.08 ± 3.87 | 89.41 ± 3.14 | 96.38 ± 2.12 |
| | Mobile-Former | 91.73 ± 2.09 | 91.88 ± 4.27 | 92.24 ± 3.03 | 92.06 ± 2.81 | 95.5 ± 2.14 | 90.37 ± 2.05 | 89.99 ± 3.63 | 88.71 ± 4.04 | 89.35 ± 2.79 | 94.31 ± 2.10 |
| TripletNet | MobileViTv2 | 94.31 ± 3.43 | 94.81 ± 4.01 | 94.18 ± 3.28 | 94.5 ± 3.69 | 97.31 ± 2.06 | 90.55 ± 2.34 | 91.39 ± 2.80 | 91.13 ± 4.33 | 91.26 ± 2.55 | 96.67 ± 2.19 |
| | LeViT | 94.34 ± 5.29 | 93.80 ± 5.37 | 93.95 ± 2.31 | 93.87 ± 5.33 | 96.67 ± 2.79 | 90.68 ± 4.09 | 89.99 ± 2.85 | 90.71 ± 2.58 | 90.35 ± 3.36 | 95.71 ± 5.60 |
| | Mobile-Former | 94.98 ± 3.18 | 94.65 ± 5.80 | 95.48 ± 2.37 | 95.06 ± 4.11 | 97.43 ± 2.57 | 92.38 ± 3.15 | 92.66 ± 2.86 | 93.08 ± 4.15 | 92.87 ± 3.00 | 96.94 ± 2.04 |
| ProtoNet | MobileViTv2 | 95.31 ± 4.54 | 95.15 ± 4.95 | 96.06 ± 2.77 | 95.6 ± 4.73 | 97.81 ± 3.18 | 91.5 ± 2.67 | 92.34 ± 2.80 | 94.88 ± 4.07 | 93.59 ± 2.73 | 96.2 ± 2.06 |
| | LeViT | 95.62 ± 2.08 | 95.61 ± 3.79 | 94.96 ± 3.08 | 95.29 ± 2.68 | 98.21 ± 5.58 | 93.91 ± 3.09 | 93.85 ± 5.96 | 92.6 ± 4.43 | 93.22 ± 4.07 | 98.13 ± 4.05 |
| | Mobile-Former | 95.60 ± 4.33 | 94.61 ± 3.19 | 94.49 ± 2.43 | 94.55 ± 3.68 | 96.98 ± 2.73 | 94.42 ± 2.45 | 92.42 ± 2.75 | 92.19 ± 5.59 | 92.3 ± 2.59 | 95.02 ± 2.47 |
| MatchingNet | MobileViTv2 | 95.91 ± 4.76 | 96.25 ± 4.42 | 96.19 ± 4.49 | 96.22 ± 4.58 | 97.88 ± 3.96 | 94 ± 5.52 | 95.2 ± 2.27 | 94.87 ± 4.86 | 95.03 ± 3.22 | 96.98 ± 3.56 |
| | LeViT | 95.86 ± 4.63 | 96.26 ± 4.94 | 96.68 ± 4.42 | 96.47 ± 4.78 | 98.84 ± 3.50 | 94.49 ± 5.16 | 94.4 ± 5.78 | 93.06 ± 5.91 | 93.72 ± 5.46 | 98.27 ± 2.62 |
| | Mobile-Former | 96.88 ± 2.87 | 96.27 ± 3.45 | 95.93 ± 2.11 | 96.1 ± 3.13 | 97.4 ± 3.90 | 95.22 ± 4.17 | 94.07 ± 2.90 | 92.28 ± 3.55 | 93.17 ± 3.42 | 96.15 ± 2.55 |
| MAML | MobileViTv2 | 96.15 ± 4.16 | 96.48 ± 3.86 | 95.5 ± 2.27 | 95.99 ± 4.00 | 98.16 ± 5.33 | 93.14 ± 5.19 | 92.61 ± 2.70 | 93.71 ± 3.15 | 93.16 ± 3.55 | 95.51 ± 2.46 |
| | LeViT | 96.69 ± 5.63 | 97.46 ± 5.08 | 97.27 ± 4.16 | 97.37 ± 5.34 | 98.36 ± 2.13 | 95.1 ± 3.86 | 95.05 ± 3.36 | 95.32 ± 3.28 | 95.19 ± 3.59 | 97.26 ± 3.63 |
| | Mobile-Former | 96.55 ± 2.15 | 95.96 ± 5.57 | 96.33 ± 4.79 | 96.15 ± 3.10 | 98.5 ± 2.38 | 94.58 ± 4.52 | 93.74 ± 2.86 | 93.38 ± 2.68 | 93.56 ± 3.51 | 97.04 ± 3.70 |
| RelationNet | MobileViTv2 | 94.93 ± 5.13 | 94.69 ± 5.16 | 94.11 ± 5.09 | 94.4 ± 5.14 | 97.65 ± 5.69 | 92.46 ± 3.62 | 91.51 ± 2.27 | 92.71 ± 5.74 | 92.1 ± 2.79 | 94.94 ± 3.75 |
| | LeViT | 94.00 ± 2.19 | 93.18 ± 2.46 | 92.59 ± 2.19 | 92.89 ± 2.31 | 98.54 ± 2.50 | 91.06 ± 2.54 | 91.02 ± 3.99 | 91.12 ± 5.42 | 91.07 ± 3.10 | 95.69 ± 4.95 |
| | Mobile-Former | 95.32 ± 5.95 | 95.36 ± 2.69 | 95.39 ± 5.81 | 95.37 ± 3.70 | 96.82 ± 2.14 | 91.91 ± 5.45 | 91.96 ± 4.21 | 93.1 ± 4.48 | 92.52 ± 4.75 | 96.46 ± 2.49 |
| Proposed | | 98.51 ± 2.09 | 98.92 ± 2.65 | 98.31 ± 3.31 | 98.61 ± 3.49 | 99.63 ± 5.67 | 96.13 ± 2.17 | 94.22 ± 2.35 | 94.55 ± 2.39 | 94.35 ± 2.26 | 99.31 ± 2.47 |

**Table 4.** Comparison of numerical results of the proposed model against baselines on leaf dataset in the 5-way 1-shot setting.

| FSL Method | Backbone | Biotic Stress Classification | | | | | Severity Estimation | | | | |
|---|---|---|---|---|---|---|---|---|---|---|---|
| | | Accuracy | Precision | Recall | F1-Score | AUC | Accuracy | Precision | Recall | F1-Score | AUC |
| SiameseNet | MobileViTv2 | 92.32 ± 2.98 | 91.18 ± 4.84 | 91.86 ± 2.85 | 91.52 ± 3.69 | 94.95 ± 1.58 | 87.8 ± 2.46 | 89.57 ± 3.59 | 88.81 ± 1.88 | 89.19 ± 2.92 | 94.22 ± 3.87 |
| | LeViT | 90.1 ± 1.76 | 91.46 ± 4.24 | 90.79 ± 1.10 | 91.12 ± 2.48 | 95.12 ± 3.19 | 90.28 ± 3.96 | 89.93 ± 4.04 | 90.97 ± 1.82 | 90.44 ± 4.00 | 92.9 ± 1.58 |
| | Mobile-Former | 89.48 ± 4.65 | 88.09 ± 4.80 | 88.8 ± 3.41 | 88.44 ± 4.72 | 94.06 ± 1.92 | 88.96 ± 2.64 | 87.95 ± 2.53 | 87.88 ± 4.13 | 87.91 ± 2.59 | 93.23 ± 2.59 |
| TripletNet | MobileViTv2 | 94.73 ± 1.07 | 93.18 ± 4.46 | 90.51 ± 4.96 | 91.83 ± 1.73 | 97.95 ± 3.42 | 90.85 ± 1.17 | 93.06 ± 1.58 | 91.2 ± 2.32 | 92.12 ± 1.34 | 95.21 ± 2.95 |
| | LeViT | 93.56 ± 4.28 | 92.91 ± 4.97 | 93.48 ± 1.73 | 93.2 ± 4.60 | 95.51 ± 2.58 | 92.25 ± 1.63 | 90.19 ± 2.97 | 89.34 ± 4.49 | 89.76 ± 2.11 | 94.53 ± 1.19 |
| | Mobile-Former | 91.15 ± 1.54 | 92.56 ± 4.19 | 91.77 ± 2.04 | 92.16 ± 2.25 | 95.45 ± 4.33 | 88.5 ± 1.93 | 89.14 ± 3.51 | 88.32 ± 3.79 | 88.73 ± 2.49 | 93.45 ± 2.95 |
| ProtoNet | MobileViTv2 | 93.1 ± 2.58 | 93.02 ± 1.76 | 94.04 ± 1.35 | 93.53 ± 2.09 | 96.45 ± 4.08 | 91.89 ± 3.96 | 90.44 ± 2.53 | 90.43 ± 3.61 | 90.43 ± 3.08 | 95.04 ± 3.02 |
| | LeViT | 92.01 ± 2.03 | 91.44 ± 4.87 | 92.59 ± 1.26 | 92.01 ± 2.87 | 97.04 ± 2.22 | 93.15 ± 3.50 | 90.51 ± 2.15 | 92.73 ± 1.35 | 91.61 ± 2.66 | 94.4 ± 3.96 |
| | Mobile-Former | 92.47 ± 3.83 | 94.63 ± 2.75 | 93.08 ± 3.64 | 93.85 ± 3.20 | 97.94 ± 4.95 | 91.26 ± 1.07 | 92.59 ± 2.15 | 93.71 ± 2.72 | 93.15 ± 1.42 | 96.28 ± 3.44 |
| MatchingNet | MobileViTv2 | 94.27 ± 3.41 | 92.86 ± 3.89 | 94.93 ± 3.24 | 93.89 ± 3.63 | 97.69 ± 1.70 | 91.01 ± 1.04 | 91.36 ± 3.15 | 93.32 ± 4.04 | 92.33 ± 1.56 | 97.26 ± 2.94 |
| | LeViT | 94.68 ± 2.84 | 93.11 ± 1.58 | 94.62 ± 3.67 | 93.86 ± 2.03 | 97.08 ± 1.42 | 91.19 ± 1.76 | 92.13 ± 4.60 | 91.96 ± 2.73 | 92.05 ± 2.55 | 95 ± 2.64 |
| | Mobile-Former | 94.25 ± 2.42 | 93.11 ± 2.03 | 95.48 ± 2.83 | 94.28 ± 2.20 | 97.93 ± 4.87 | 91.08 ± 4.54 | 92.58 ± 2.00 | 91.74 ± 4.61 | 92.16 ± 2.78 | 96.76 ± 2.14 |
| MAML | MobileViTv2 | 94.31 ± 2.53 | 92.98 ± 3.51 | 95.04 ± 4.66 | 94 ± 2.94 | 97.47 ± 4.89 | 92.14 ± 1.30 | 91.36 ± 4.33 | 93.19 ± 2.10 | 92.27 ± 2.00 | 96.99 ± 2.07 |
| | LeViT | 94.55 ± 3.64 | 93.29 ± 1.73 | 96.49 ± 3.91 | 94.87 ± 2.35 | 98.81 ± 1.33 | 91.8 ± 3.88 | 92.73 ± 4.98 | 92.4 ± 2.95 | 92.57 ± 4.36 | 97.14 ± 4.05 |
| | Mobile-Former | 93.3 ± 1.01 | 92.86 ± 3.91 | 92.78 ± 3.58 | 92.82 ± 1.61 | 95.88 ± 1.71 | 91.21 ± 1.21 | 91.27 ± 4.51 | 94.19 ± 2.29 | 92.7 ± 1.90 | 96.31 ± 2.09 |
| RelationNet | MobileViTv2 | 94.11 ± 4.47 | 92.05 ± 2.84 | 93.88 ± 3.11 | 92.96 ± 3.48 | 97.75 ± 3.90 | 92.36 ± 4.51 | 90.74 ± 4.85 | 91.22 ± 4.88 | 90.98 ± 4.67 | 97.28 ± 4.59 |
| | LeViT | 94.69 ± 2.57 | 94.32 ± 1.83 | 94.18 ± 4.48 | 94.25 ± 2.14 | 98.01 ± 1.37 | 90.75 ± 3.05 | 89.79 ± 2.65 | 87.27 ± 4.06 | 88.51 ± 2.83 | 96.96 ± 3.88 |
| | Mobile-Former | 92.69 ± 3.43 | 94.03 ± 3.11 | 94.61 ± 2.10 | 94.32 ± 3.26 | 97.12 ± 4.27 | 90.06 ± 1.61 | 93.11 ± 4.70 | 88.56 ± 2.19 | 90.78 ± 2.40 | 97.65 ± 1.78 |
| Proposed | | 96.72 ± 2.67 | 96.72 ± 5.76 | 96.22 ± 3.99 | 96.46 ± 2.65 | 98.89 ± 2.60 | 94.34 ± 2.02 | 94.26 ± 2.01 | 94.37 ± 2.49 | 94.24 ± 2.02 | 98.84 ± 4.74 |

To further understand and scrutinize the behavior of our model (five-way one-shot), Figure 4 provides a visualization of the class-level performance by plotting the confusion matrices for both classification and severity estimation tasks. For coffee biotic stress classification, it is worth noting that the class "Brown leaf spot" attains the lowest detection performance with 95.7% precision, while the class "healthy" attains the highest detection performance with 98.1% precision. For severity estimation, it is worth noting that the class "very high" attains the lowest detection performance with 91% precision, while the class "high" attains the highest detection performance with 95% precision. This can be attributed to the class imbalance or the in-between severity levels that cause the model to be confused.

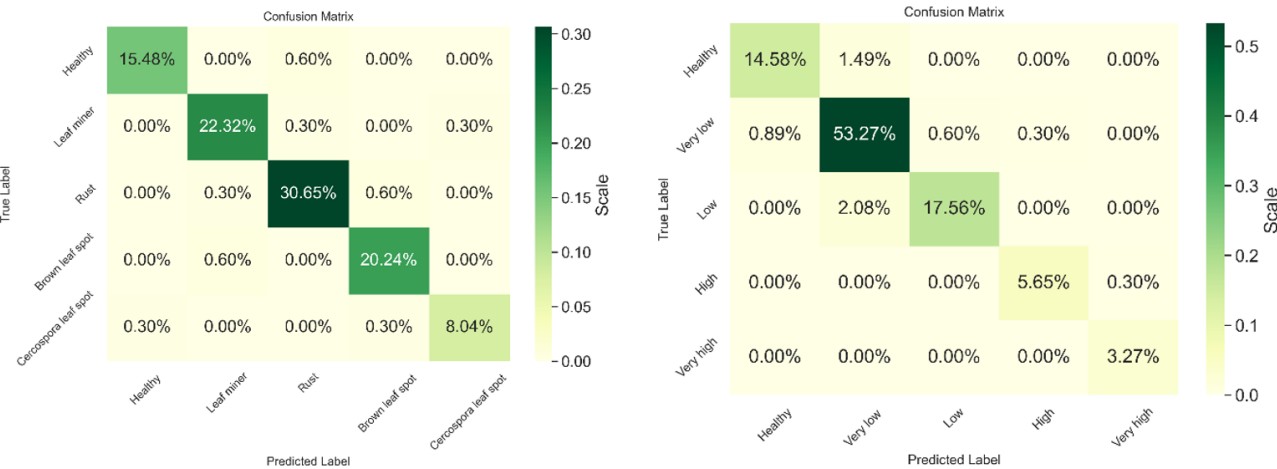

**Figure 4.** Confusion matrix of the proposed model on leaf dataset for coffee biotic stress classification (**Left**) and severity estimation (**Right**) in 5-way 1-shot setting.

In Table 5, we present the quantitative results obtained from the different methods under both five-way five-shot and five-way one-shot training on the symptom dataset. The tabulated results only represent the performance of the models on coffee biotic stress classification. It is worth noting that the NAML achieves the highest performance among other baselines, while SiameseNet gets the lowest performance. This observation applies to both five-way one-shot as well as five-way five-shot settings. Conforming to our findings on the leaf dataset, the proposed model shows competing for performance outperforming overall baselines on the symptom dataset. Notably, the competing baselines don't show significant performance gain when increasing the number of shots. The same behavior applies to the proposed model, in which the classification performance only reduces by 1% when shifting from five shots to one shot. This can be attributed to the fact that the leaf dataset encompasses only a few categories, which makes the learner not require many support instances per category to achieve good generalization.

In Figure 5, we present the confusion matrix of the proposed model on the symptom dataset. For five-way five-shot settings, it is observable that the proposed model recognizes different types of biotic stresses with an average precision above 98%. For the five-way one-shot setting, it is worth noting that the class "Brown leaf spot" attains the lowest detection performance with an average precision of 94.8%, while the class "Leaf miner" attains the highest detection performance with an average precision of 99.1%. This is explained by the fact that the variation in the representativeness of support symptom samples has a significant impact on the class-level performance and overall model performance.

**Table 5.** Comparison of numerical results of the proposed model against baselines on symptom dataset.

| FSL Method | Backbone | Biotic Stress Classification (5-Way 5-Shot) | | | | | Biotic Stress Classification (5-Way 1-Shot) | | | | |
|---|---|---|---|---|---|---|---|---|---|---|---|
| | | Accuracy | Precision | Recall | F1-Score | AUC | Accuracy | Precision | Recall | F1-Score | AUC |
| SiameseNet | MobileViTv2 | 92.34 ± 2.64 | 92.88 ± 3.62 | 92.48 ± 5.62 | 92.68 ± 3.05 | 95.11 ± 4.31 | 92.24 ± 3.87 | 93.24 ± 2.33 | 91.81 ± 3.41 | 92.52 ± 2.91 | 95.46 ± 2.07 |
| | LeViT | 91.99 ± 2.78 | 90.99 ± 2.61 | 91.3 ± 2.56 | 91.14 ± 2.69 | 96.55 ± 2.28 | 92.36 ± 3.64 | 90.47 ± 3.07 | 90.38 ± 4.25 | 90.42 ± 3.33 | 97.08 ± 4.85 |
| | Mobile-Former | 91.23 ± 5.51 | 91.38 ± 3.78 | 92.24 ± 5.09 | 91.81 ± 4.49 | 94.5 ± 2.82 | 91.61 ± 5.44 | 91.04 ± 4.36 | 92.46 ± 3.63 | 91.74 ± 4.84 | 94.47 ± 3.70 |
| TripletNet | MobileViTv2 | 93.31 ± 2.39 | 93.81 ± 2.38 | 94.18 ± 2.24 | 93.99 ± 2.39 | 97.31 ± 5.57 | 93.33 ± 5.60 | 94.32 ± 4.78 | 93.99 ± 5.88 | 94.15 ± 5.16 | 96.81 ± 5.11 |
| | LeViT | 93.84 ± 3.62 | 93.8 ± 3.83 | 92.95 ± 3.95 | 93.37 ± 3.72 | 96.17 ± 3.59 | 93.83 ± 2.49 | 94.52 ± 2.53 | 93.31 ± 3.29 | 93.92 ± 2.51 | 95.83 ± 5.85 |
| | Mobile-Former | 94.98 ± 5.22 | 93.65 ± 3.74 | 95.48 ± 4.44 | 94.56 ± 4.36 | 96.93 ± 3.23 | 94.91 ± 4.34 | 94.45 ± 4.30 | 94.5 ± 5.28 | 94.48 ± 4.32 | 97.07 ± 3.05 |
| ProtoNet | MobileViTv2 | 95.31 ± 4.10 | 94.15 ± 3.94 | 95.56 ± 3.82 | 94.85 ± 4.02 | 96.81 ± 4.34 | 95.69 ± 3.34 | 93.5 ± 2.51 | 94.83 ± 4.66 | 94.16 ± 2.87 | 96.25 ± 2.65 |
| | LeViT | 95.31 ± 5.06 | 95.61 ± 3.91 | 94.46 ± 2.08 | 95.03 ± 4.41 | 98.21 ± 4.80 | 95.09 ± 4.47 | 95.84 ± 5.18 | 95.4 ± 2.30 | 95.62 ± 4.80 | 97.27 ± 3.40 |
| | Mobile-Former | 94.6 ± 5.44 | 94.61 ± 3.28 | 94.49 ± 3.45 | 94.55 ± 4.09 | 96.48 ± 2.03 | 94.75 ± 2.98 | 94.54 ± 4.93 | 93.69 ± 2.30 | 94.11 ± 3.71 | 96.91 ± 4.15 |
| MatchingNet | MobileViTv2 | 95.91 ± 4.70 | 95.25 ± 5.46 | 95.19 ± 2.19 | 95.22 ± 5.05 | 97.88 ± 3.71 | 95.57 ± 3.36 | 95.31 ± 4.74 | 94.41 ± 4.31 | 94.86 ± 3.93 | 98.12 ± 4.55 |
| | LeViT | 95.86 ± 3.55 | 95.76 ± 2.39 | 95.68 ± 2.39 | 95.72 ± 2.86 | 97.84 ± 2.04 | 95.99 ± 4.38 | 95.83 ± 4.14 | 94.92 ± 5.54 | 95.37 ± 4.26 | 97.24 ± 5.43 |
| | Mobile-Former | 95.88 ± 3.50 | 95.27 ± 5.32 | 94.93 ± 5.39 | 95.1 ± 4.22 | 96.4 ± 4.72 | 96.81 ± 2.87 | 95.18 ± 2.41 | 95.05 ± 3.24 | 95.12 ± 2.62 | 96.99 ± 4.74 |
| MAML | MobileViTv2 | 96.15 ± 5.19 | 95.98 ± 3.98 | 95.5 ± 4.35 | 95.74 ± 4.51 | 97.16 ± 5.77 | 96.83 ± 4.58 | 95.89 ± 2.83 | 95.03 ± 5.64 | 95.46 ± 3.50 | 96.87 ± 4.22 |
| | LeViT | 95.89 ± 4.07 | 96.46 ± 2.77 | 96.77 ± 3.73 | 96.61 ± 3.29 | 97.36 ± 2.62 | 95.02 ± 3.58 | 95.82 ± 3.16 | 97.57 ± 4.07 | 96.69 ± 3.36 | 97.83 ± 4.01 |
| | Mobile-Former | 96.05 ± 3.81 | 94.96 ± 2.19 | 96.33 ± 4.43 | 95.64 ± 2.78 | 98.5 ± 3.39 | 95.96 ± 5.30 | 94.66 ± 2.88 | 96.56 ± 4.61 | 95.6 ± 3.73 | 99.3 ± 2.18 |
| RelationNet | MobileViTv2 | 94.93 ± 5.86 | 93.69 ± 2.04 | 93.11 ± 3.41 | 93.4 ± 3.03 | 97.65 ± 4.51 | 95.92 ± 4.29 | 94.37 ± 3.34 | 92.24 ± 4.56 | 93.29 ± 3.76 | 97.02 ± 3.84 |
| | LeViT | 93.5 ± 4.75 | 92.18 ± 5.13 | 91.59 ± 3.75 | 91.88 ± 4.93 | 97.54 ± 3.46 | 93.58 ± 2.79 | 91.75 ± 5.20 | 90.75 ± 4.46 | 91.25 ± 3.63 | 97.97 ± 3.41 |
| | Mobile-Former | 94.32 ± 2.50 | 95.36 ± 4.84 | 94.39 ± 5.92 | 94.87 ± 3.29 | 96.32 ± 3.89 | 93.33 ± 2.48 | 95.7 ± 3.53 | 93.4 ± 3.24 | 94.54 ± 2.92 | 97.19 ± 5.40 |
| Proposed | | 98.34 ± 1.29 | 98.4 ± 3.14 | 98.14 ± 2.23 | 98.27 ± 1.83 | 99.68 ± 2.35 | 97.79 ± 1.18 | 97.22 ± 1.47 | 97.38 ± 1.67 | 97.30 ± 1.31 | 99.12 ± 3.93 |

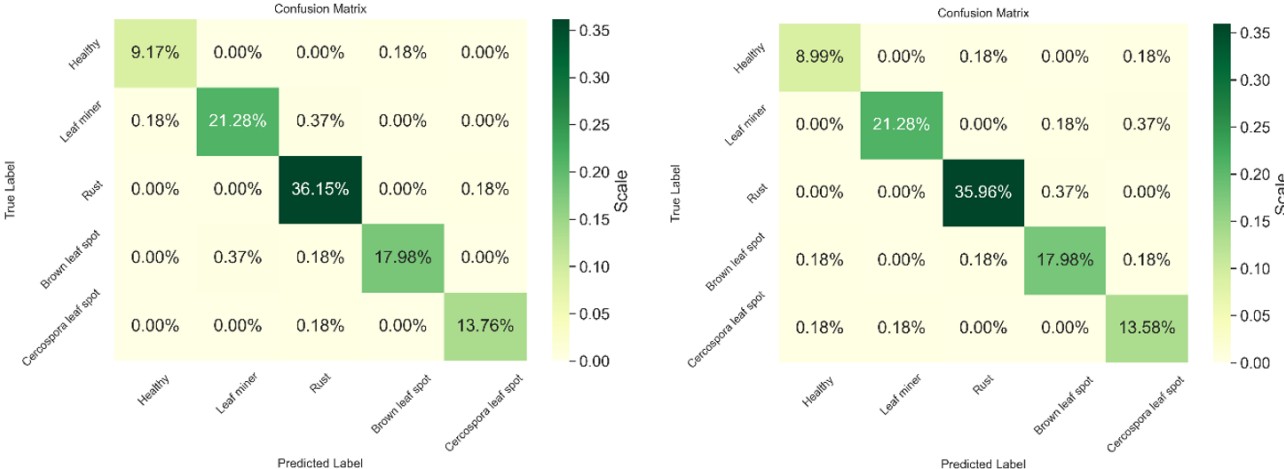

**Figure 5.** Confusion matrix of the proposed model on symptom dataset in 5-way 5-shot setting (**Left**) and in 5-way 1-shot setting (**Right**).

In Table 6, we display the results of two-tailed t-test experiments applied to assess and statistically analyze the role significance of achieved improvements under both five-way five-shot and five-way one-shot settings. In these experiments, we use a threshold value of 0.05. As observed, the statistical results reveal the remarkable classification accuracy in categorizing coffee leaf diseases and determining their severity level. The statistical significance is observed for both types of settings validating the robustness of our model and its potential to address the complications of multi-task leaf diagnosis efficiently. Based on these outcomes, we can conclude that the performance improvements achieved by our approach are realistic and not achieved by chance, which proves the competitive advantage of our mode over the competing models.

**Table 6.** Statistical Analysis Results for 5-Way 5-Shot and 5-Way 1-Shot Settings.

| | 5-Way 5-Shot | | 5-Way 1-Shot | |
|---|---|---|---|---|
| | **Biotic Stress Classification** | **Severity Estimation** | **Biotic Stress Classification** | **Severity Estimation** |
| Proposed vs. SiameseNet | $3.983 \times 10^{-11}$ | $8.405 \times 10^{-11}$ | $8.014 \times 10^{-15}$ | $2.377 \times 10^{-70}$ |
| Proposed vs. TripletNet | $3.399 \times 10^{-40}$ | $4.268 \times 10^{-14}$ | $7.375 \times 10^{-80}$ | $1.662 \times 10^{-90}$ |
| Proposed vs. ProtoNet | $8.185 \times 10^{-20}$ | $1.363 \times 10^{-21}$ | $8.313 \times 10^{-90}$ | $6.254 \times 10^{-40}$ |
| Proposed vs. MatchingNet | $4.482 \times 10^{-11}$ | $6.694 \times 10^{-31}$ | $8.539 \times 10^{-12}$ | $2.501 \times 10^{-50}$ |
| Proposed vs. MAML | $6.641 \times 10^{-50}$ | $8.805 \times 10^{-12}$ | $4.539 \times 10^{-13}$ | $3.357 \times 10^{-80}$ |
| Proposed vs. RelationNet | $4.369 \times 10^{-3}$ | $6.220 \times 10^{-8}$ | $5.713 \times 10^{-7}$ | $2.819 \times 10^{-11}$ |

## 5.2. Ablation Analysis

In this section, the results from ablation experiments are discussed to explain the contribution of different building blocks to the final performance of the proposed model. In this context, a set of learner ablation experiments is implemented to compare the performance of the MVT to other multi-path learners from the literature. In Figure 6, the results of these experiments on the leaf dataset are reported for both five-way five-shot and five-way one-shot settings. It is worth noting that the proposed MVT enables our model to achieve higher performance on biotic stress classification and severity estimation tasks. The findings further demonstrate the efficiency of MVT. This can be attributed to the ability of MVT to extract the fine-grained features from both support and query images while considering the feature interaction among them.

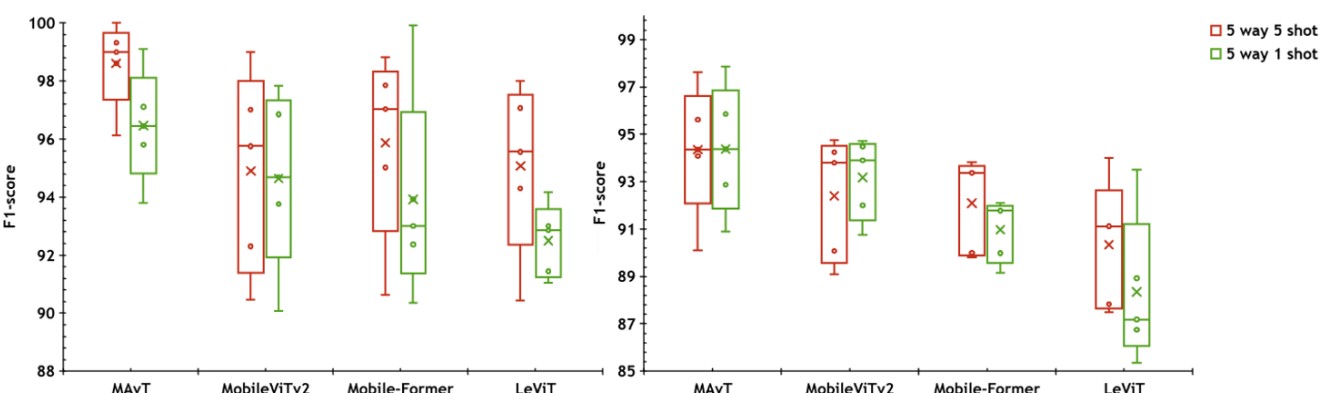

**Figure 6.** Results of learner ablation experiments on leaf dataset for coffee biotic stress classification (**Left**) and severity estimation (**Right**).

Moreover, to interpret the performance of our MVT, a comprehensive ablation analysis is conducted to evaluate the efficacy of our IPE against the conventional PE, as shown in Figure 7. This experimental analysis is expected to elucidate the distinct contributions of IPE to the overall model performance. As shown, the detection performance improvements achieved by our IPE module, demonstrated its effectiveness in modeling complex spatial relations and context representations within leaf images. Amazingly, the results on the severity estimation task conform with the findings on the detection task, which further supports our claims regarding the role of the IPE module. These findings collectively lay the groundwork for future research intended to enhance embedding strategies in the context of multi-task leaf diagnosis, promoting an in-depth interpretation of the complicated associations between spatial information and accurate disease classification.

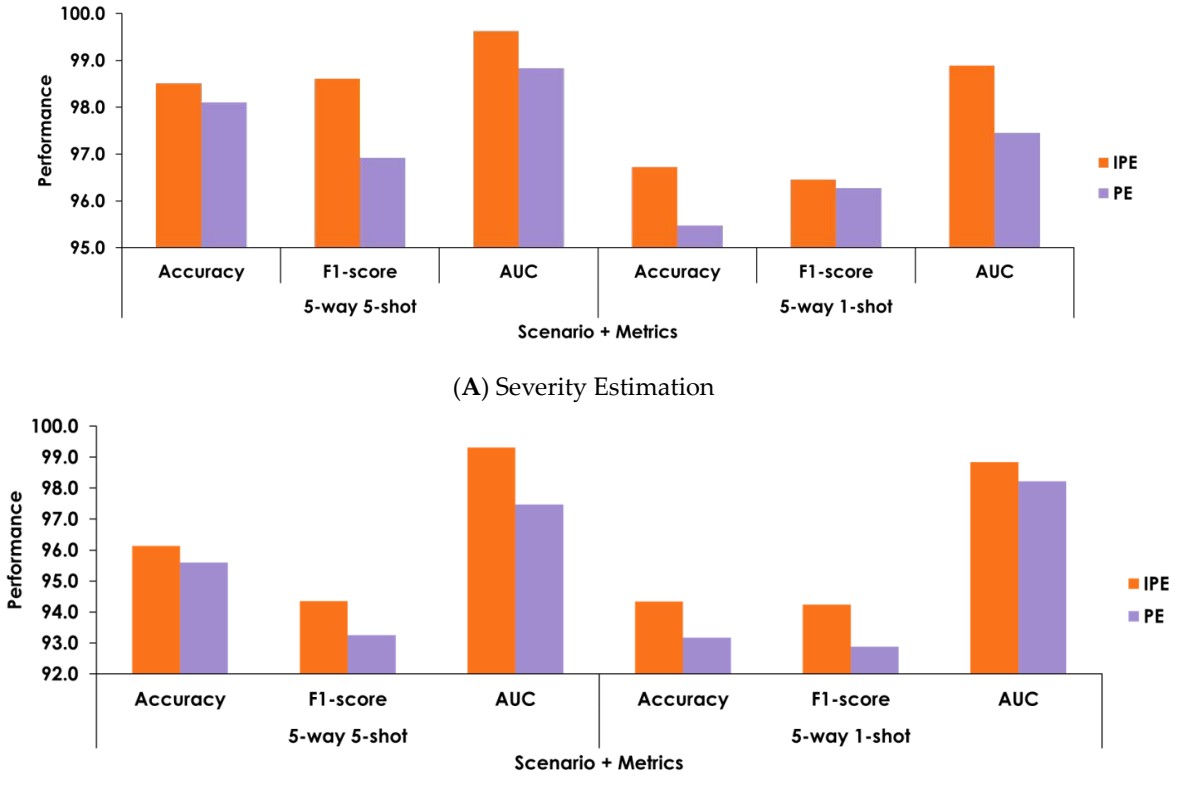

(**A**) Severity Estimation

(**B**) Biotic stress Classification

**Figure 7.** Ablation Analysis of Model Performance with IPE vs. Standard PE.

According to the above findings, it is worth noting that our work has important ramifications for the social and economic fabric of rural areas in addition to their experimental merits. The proposed framework has the potential to completely alter the coffee farming environment, especially in areas where the industry is vital to the economy. With the quick and accurate multi-task diagnosis of coffee leaf diseases, the proposed model can provide farmers with the data they need to take corrective and preventative steps. The consequence is improved food security for populations that rely heavily on coffee for both economic and nutritional support. The financial security of farmers is bolstered by higher yields, and their ability to weather agricultural crises is strengthened [30–32]. Our findings support sustainable agriculture's broader mission to improve the lives of farmers and rural communities while also bolstering regional economies and ensuring a reliable supply of staple food. Though the origins of our study are in the fields of machine learning and multi-task leaf diagnostics, its implications are substantially more general. The potential of our model to solve real-world difficulties faced by coffee farmers increases the program's potential social and economic impact. The capacity to correctly identify biotic stress affecting coffee leaves is vital not just for effective disease management but also for the continued existence of agricultural communities. Since correct diagnosis improves harvests, it helps ensure that people in coffee-growing areas have access to a safe and sufficient food supply. As a result of increasing yields and farmer incomes, the economy stands a better chance of remaining stable thanks to this productivity boost. Our novel strategy has contributed to a robust and sustainable agriculture sector [33–36].

## 6. Conclusions

This work presents a novel meta-learning approach for a few-shot multi-task diagnosis of biotic stresses from coffee leaf images. A new MVT network is proposed to perform mixed contextual attention on both query and support images of the meta-training dataset. An adaptive meta-training strategy is introduced based on an objective function that acclimates to each episode according to its state throughout the inner-loop optimization, hence empowering the model's generalizability. The experimental results demonstrated that the proposed targeted approach not only enhances the diagnosis and management of coffee leaves but also empowers coffee farmers with efficient and accurate tools to combat diseases, safeguard crop productivity, and promote sustainable practices. The proposed holistic approach enables site-specific disease management policies tailored to the exceptional conditions of each coffee plantation. However, the centralized scenario of our model remains a significant limit to the use and deployment of our model in real-world smart farms which are geographically distributed across different locations.

In line with continuous advancements in the field of multi-task coffee leaf diagnosis, many promising avenues are envisioned for future exploration. Firstly, our knowledgeable meta-learning approach can be fine-tuned to optimize its adaptability to a variety of crop diseases and varied environmental settings. In addition, the continuous evolution of sustainable agriculture highlights the importance of integrating real-time monitoring and IoT technologies into the proposed approach to allow timely and data-driven decision-making for farmers. The potential for our investigation to grow to incorporate remote sensing data and satellite pictures for disease forecasting and early diagnosis is quite encouraging. We also plan to work with agronomists and ecologists to better understand the intricate interplay between crop health and environmental factors through the development of interdisciplinary collaborations. Furthermore, Figure 8 displays the comprehensive ablation analysis that thoroughly assesses the performance of our model when implemented with the proposed adaptive meta-training, and when implemented with the standard meta-training approach. As shown, the experimental results show that adaptive meta-training reveals a noteworthy advantage and demonstrates superior adaptability to developing task distributions and complexities. With the dynamic adjustment ability of our meta-learner, our approach excels in capturing task-specific nuances, thus showcasing higher accuracy and robustness in multi-task leaf diagnosis. On the other hand, with the conventional

meta-training approach, our model suffers from limited adaptability in the face of changing circumstances.

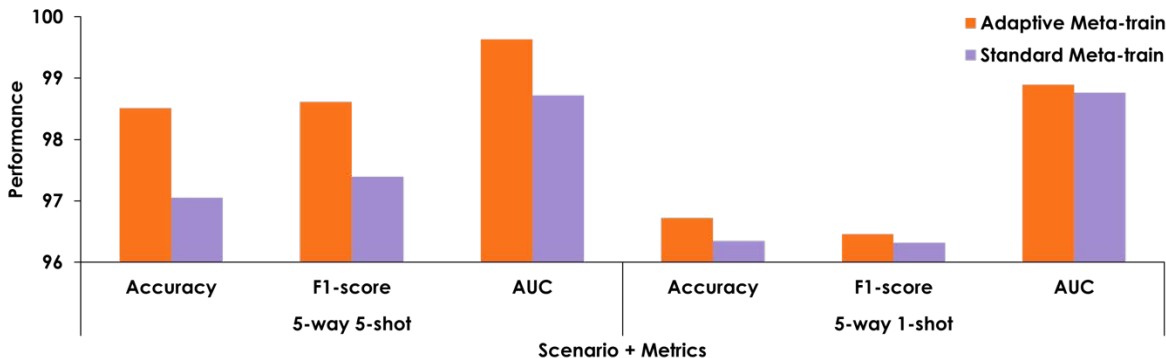

(**A**) Severity Estimation

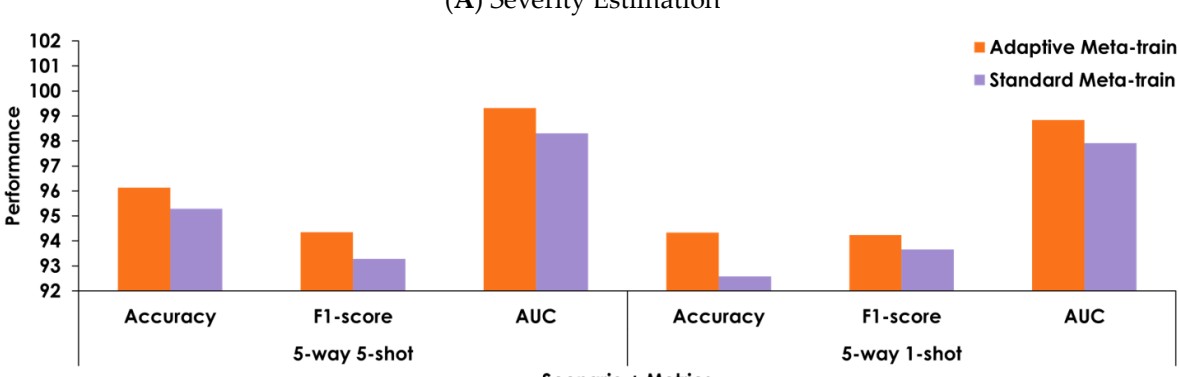

(**B**) Biotic Stress Classification

**Figure 8.** Ablation Analysis of Model Performance with Adaptive Meta-training vs. Standard Meta-training.

**Author Contributions:** Methodology, A.A.S. and W.T.A.-N.; Software, A.A.S. and W.T.A.-N.; Validation, A.A.S. and W.T.A.-N.; Investigation, A.A.S.; Data curation, A.A.S.; Writing—original draft, A.A.S.; Writing—review and editing, W.T.A.-N.; Supervision, A.A.S. All authors have read and agreed to the published version of the manuscript.

**Funding:** This research received no external funding.

**Institutional Review Board Statement:** Not applicable.

**Informed Consent Statement:** Not applicable.

**Data Availability Statement:** The datasets generated during and/or analyzed during the current study are not publicly available due to the privacy-preserving nature of the data but are available from the corresponding author upon reasonable request.

**Acknowledgments:** The authors extend their appreciation to the Deputyship for Research & Innovation, Ministry of Education, in Saudi Arabia for funding this research work through the project number ISP22-35.

**Conflicts of Interest:** The authors declare no conflict of interest.

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
