# Peer review of "Sustainable Coffee Leaf Diagnosis: A Deep Knowledgeable Meta-Learning Approach"

_sustainability, doi:10.3390/su152416791_

Round 1

Reviewer 1 Report

The paper presents an original approach by combining multi-task meta-learning, mixed contextual attention, and knowledge distillation for biotic stress classification, addressing the limitations of existing methods. The study addresses a pertinent issue in agriculture, offering potential advancements in coffee leaf disease diagnosis and management, aligning with the demands of sustainable agriculture. The paper conducts exhaustive experiments, yielding competitive results that highlight the efficacy of the proposed method, making the findings more convincing and impactful. However, some comments can further improve the quality of the paper.

1.       In the abstract, the authors named their model “A mixed Agri-vision transformer (MAvT) learner”. The authors are suggested to carefully reconsider the naming of the proposed model to further improve the readability of the manuscript.

2.       All terms in the manuscript should also be carefully revised. For example, the term “agri-vision” in section 1.1. and 1.2. Do you mean computer vision?

3.       To align the paper with the sustainability domain, consider explicitly mentioning how the proposed multi-task meta-learning framework contributes to sustainable agriculture practices. The abstract section should highlight how the approach could potentially reduce the environmental impact associated with conventional diagnostic methods, thus promoting sustainability in coffee leaf disease management.

4.       In the introduction section, expand on the motivation behind integrating sustainability principles into the study. Explain why sustainable approaches are critical in addressing the challenges of crop health monitoring and disease management in agriculture.

5.       In the introduction section, clearly state how the multi-task meta-learning framework contributes to resource efficiency and emphasize the potential for using fewer training samples and computational resources, which aligns with sustainability objectives.

6.       While the related work section provides an insightful overview of the existing literature, it would be beneficial for the authors to explicitly emphasize the primary novelty of their proposed multi-task meta-learning framework for coffee leaf disease diagnosis.

7.       In the related work section, If the proposed framework's focus on few-shot classification is a key departure from prior methods, emphasize how this contributes to resource efficiency and adaptability to new challenges in agricultural settings.

8.       In section 3, the authors claimed “The design of distillation loss also considered Kullback–Leibler divergence and logits distillation, but our model achieves higher performance with JS distillation. To this end, JS distillation is selected between learners of the previous episodes and the learner of the current episode as a penalty to impose their likeness.”. Ablation experiments should be extended to validate this claim.

9.       In section 3,  the authors stated that “the Born-again [20] approach is used to put on JS successively to engender manifold generations…”. I think this part needs to be extended with more discussions to justify this design choice.

10.    Beyond the comparative experiments, the authors are suggested to extend the experimental section to validate the efficiency of “Adaptive Meta-training” over traditional meta-training.

11.    A thorough proofreading is recommended to eliminate occasional grammatical errors and ensure the manuscript's language adheres to the highest standards.

12.    In section 4.4, expand Table 2 with more details about the training hyper-parameters to further improve the reproducibility. Also, check the word “benchmark” in the caption of Table 2.

13.    In section 5.1, the authors presented comprehensive comparative experimental results in Tables 3-5. Despite the numerical achievements of the proposed solution, statistical significance experiments are required to validate the competitive advantage over the competing solutions.

14.    The Results and Discussions section, addresses the socio-economic aspect by discussing how accurate disease diagnosis can lead to higher crop yields, thus contributing to food security and economic stability in agricultural communities.

15.    In the Results and Discussions section, the captions of the confusion matrix should be carefully revised.

16.    In the experimental part, If the novel approach exhibits superior generalizability to unseen data compared to previous methods, this is a significant point of distinction that deserves special attention.

17.    More experiments can be added to further improve the generalization ability of the proposed performance.

18.    In Algorithm 1, the authors presented Interactive Patch Embedding (IPE). However, this part of the framework misses experimental validation. Such an experiment is favorable to further improve the validity of your work.

19.    Ensure that symbols are used consistently throughout the manuscript and that their definitions are accurate and align with their usage in equations, figures, and text.

20.    Include a list of all abbreviations used in the paper, along with their full forms, in the early sections of the manuscript (e.g., after the abstract or introduction). This helps readers understand the meaning of abbreviations without having to backtrack in the text.

 Minor editing of English language required

Reviewer 2 Report

The introduction needs to be rewritten properly and the contribution of this work added clearly inside the section.

The methodology section is not okay; add a proper flowchart to show the proper steps of this study. With proper explanation, I need to add

Add discussion sections, compare your work with some published work, and cite recent papers like,

https://www.sciencedirect.com/science/article/pii/S2772442522000521

https://www.mdpi.com/2079-3197/10/10/177

Authors should fix grammar issues before resubmitting. And the coherence of the writing of this paper is not okay at all, need to fix before submitting. 

Reviewer 3 Report

Overall the paper is well written, however, there are some minor suggestions which may be incorporated to improve the quality of the manuscript.

1. Title of the paper is too long, it should be precise.

2. Section 2 and 3 should be merged in section 1 i.e; Introduction.

3. Important details of the proposed ViT like hyperparameters and trainable parameters need to be incorporated.

4. Comparison with state of the art is missing, further, it is also unclear whether other studies have used the same datasets of authors or not.

5. Future work is missing in the conclusion section.

Minor English language editing is required.

Reviewer 4 Report

The article covers a very interesting topic. The way it is presented needs some corrections:

First of all, the text should be formatted according to the Journal's guidelines.

A summary of the most important advantages and disadvantages of the methods used so far should be provided in Chapter 2 - this could be in the form of a table.

There is a lack of reference to Figure 1 in the text. there is also a lack of detailed description of the information contained therein. There is no point in inserting figures without discussing them in detail. Note also applies to other figures and tables.

Describe all the symbols used in the individual formulae.

In the conclusion, show the advantages and limitations of the proposed method, in relation to other methods.

Round 2

Reviewer 2 Report

This paper can be accepted.  Cite some recent works;

i e: 

https://link.springer.com/article/10.1007/s44230-023-00039-x

and more

Need to improve writing quality of introduction section.  It's very important 

Author Response

Comment 1: This paper can be accepted. Cite some recent works;  i e: https://link.springer.com/article/10.1007/s44230-023-00039-x. and more

Response: We are pleased to receive your positive feedback and your suggestion to cite additional recent works, including the reference you provided (https://link.springer.com/article/10.1007/s44230-023-00039-x). We have carefully reviewed your comment and have made the following revisions to the manuscript:      We have included the reference you suggested (https://link.springer.com/article/10.1007/s44230-023-00039-x) in the relevant section of our paper, and we have cited it appropriately to enhance the comprehensiveness and relevance of our literature review. Furthermore, we have conducted an extensive literature search to identify other recent works related to our topic. We have incorporated additional references from the last few years to ensure that our paper reflects the most up-to-date research in the field.

Comment 2: Need to improve writing quality of introduction section. It's very important.

Response: Thank you for your valuable feedback on our manuscript. We appreciate your constructive comment regarding the writing quality of the introduction section, recognizing its critical importance. We have carefully considered your suggestion and have taken the following steps to enhance the quality of the introduction. 1) Clarity and Conciseness; 2) Engagement; 3) Flow and Transitions; 4)  Language Quality,  and  4) Incorporation of Key Concepts.

Reviewer 4 Report

The authors have taken into account the comments made in the first review, and as a result, I believe that the article, having previously formatted the text according to the guidelines, can be published in the journal.

Author Response

Reviewer4

The authors have considered the comments made in the first review, and as a result, I believe that the article, having previously formatted the text according to the guidelines, can be published in the journal.

Response : We are delighted to receive your positive feedback and your assessment that our manuscript is now suitable for publication in the journal. We sincerely appreciate your thorough review of our work, and we are pleased that the revisions we made in response to the previous comments have met your expectations.